# NaturalBench: Evaluating Vision-Language Models on Natural Adversarial Samples

**Baiqi Li**[1][*]    **Zhiqiu Lin**[1][*]    **Wenxuan Peng**[1][*]    **Jean de Dieu Nyandwi**[1][*]    **Daniel Jiang**[1]
**Zixian Ma**[2]    **Simran Khanuja**[1]    **Ranjay Krishna**[2][†]    **Graham Neubig**[1][†]    **Deva Ramanan**[1][†]
[1]Carnegie Mellon University    [2]University of Washington

## Abstract

Vision-language models (VLMs) have made significant progress in recent visual-question-answering (VQA) benchmarks that evaluate complex visio-linguistic reasoning. However, are these models truly effective? In this work, we show that VLMs still struggle with natural images and questions that humans can easily answer, which we term **natural adversarial samples**. We also find it surprisingly easy to generate these VQA samples from natural image-text corpora using off-the-shelf models like CLIP and ChatGPT. We propose a semi-automated approach to collect a new benchmark, **NaturalBench**, for reliably evaluating VLMs with 10,000 human-verified VQA samples. Crucially, we adopt a **vision-centric** design by pairing each question with two images that yield different answers, preventing "blind" solutions from answering without using the images. This makes Natural-Bench more challenging than previous benchmarks that can largely be solved with language priors like commonsense knowledge. We evaluate *53* state-of-the-art VLMs on NaturalBench, showing that models like BLIP-3, LLaVA-OneVision, Cambrian-1, InternLM-XC2, Llama3.2-Vision, Molmo, Qwen2-VL, and even the (closed-source) GPT-4o lag 50%-70% behind human performance (which is above 90%). We analyze why NaturalBench is hard from two angles: (1) **Compositionality:** Solving NaturalBench requires diverse visio-linguistic skills, including understanding attribute bindings, object relationships, and advanced reasoning like logic and counting. To this end, unlike prior work that uses a single tag per sample, we tag each NaturalBench sample with 1 to 8 skill tags for fine-grained evaluation. (2) **Biases:** NaturalBench exposes severe biases in VLMs, as models often choose the same answer regardless of the image. We show that debiasing can be crucial for VLM performance. Lastly, we apply our benchmark curation method to diverse data sources, including long captions (over 100 words) and non-English languages like Chinese and Hindi, highlighting its potential for dynamic evaluations of VLMs.

## 1 Introduction

Recent vision-language models (VLMs) such as GPT-4o [61], GPT-4Vision [60], BLIP-3 (XGen-MM) [79], LLaVA-OneVision [37], InternLM-XC2 [14], Llama3.2-Vision [17], Molmo [10] and Qwen2-VL [76] have markedly improved performance on challenging visual-question-answering (VQA) benchmarks like MMMU [83] and MME [18]. These benchmarks evaluate VLMs across various domains, such as college-level subjects [53, 83], commonsense reasoning [18, 34], diagram comprehension [30, 52], and complex problem-solving in mathematics, coding, physics, and temporal forecasting [18, 50, 54]. Despite their progress, our research identifies a significant gap: *these models still struggle with seemingly simple questions about natural images.* Figure 1 shows such VQA samples that humans find easy to solve, while even the state-of-the-art models fail. We term these **natural adversarial samples** [24] for VLMs.

---

[*]Co-first authors; [†]Co-senior authors. Datasets and code at https://linzhiqiu.github.io/papers/naturalbench.

38th Conference on Neural Information Processing Systems (NeurIPS 2024) Track on Datasets and Benchmarks.

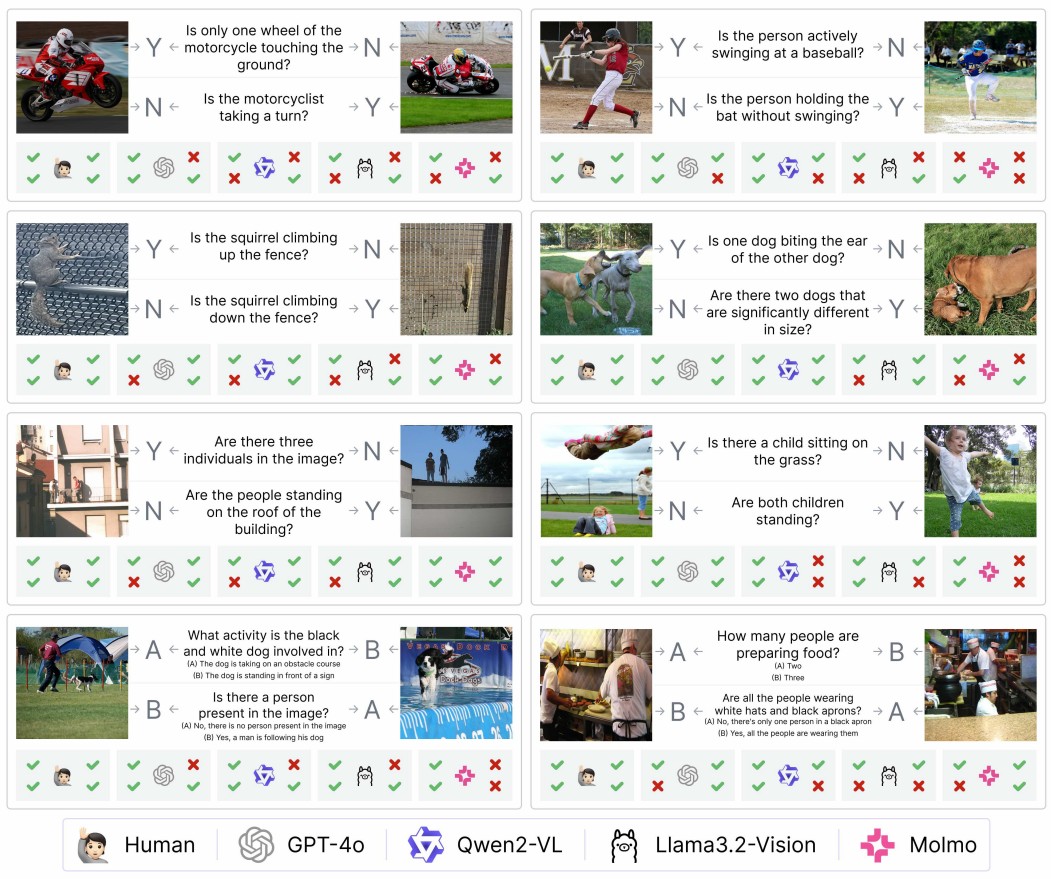

Figure 1: **NaturalBench examples** consist of two questions and two images with alternating answers to prevent "blind" models from scoring well (e.g., those that predict the same answer regardless of the image or question, as discussed in Section 3). We compare the ground-truth answer for each (image, question) pair with predictions from leading VLMs including GPT-4o (`gpt-4o-2024-08-06`), Qwen2-VL (72B), Llama3.2-Vision (90B), and Molmo (72B) (see Section 4). Even the best models like GPT-4o lags far behind human performance (which is above 90%). Figure 2 shows the pipeline for collecting these natural adversarial examples.

**Collecting natural adversarial samples.** In contrast to previous benchmarks that challenge VLMs with carefully-curated VQA samples [5, 18, 50, 73], we propose a semi-automated method to minimize human efforts by generating VQA samples from existing natural image-text datasets [42, 63] (see Figure 2). First, we identify *pairs* of image-text samples that leading VLMs like CLIP [65] fail to match correctly; typically, these pairs are visually and semantically similar. After collecting these confounding pairs, we send both samples to ChatGPT [60] to generate questions that elicit different answers for the two images. We hire human annotators to remove incorrect or irrelevant question-answer (QA) pairs by examining their corresponding images. This process is notably simpler than previous adversarial VQA benchmarks [40, 68] that train annotators to write new QA pairs that fail a targeted VQA model. Nonetheless, our VQA samples pose a "natural" challenge to state-of-the-art models without specifically targeting any.

**Avoiding "blind" solutions.** Crucially, pairing each question with two images that yield different answers enforces VLMs to rely on the visual inputs. This approach contrasts with earlier benchmarks that can be (partially) addressed by blind language models [5, 45] that do not look at images. Indeed, we demonstrate that a suite of six popular VQA benchmarks [5, 18, 30, 50, 53, 83] can be largely addressed by a blind ChatGPT that exploits language biases. For instance, benchmarks like MME [18] contain questions like "*Is there a black giraffe in the image?*", which can be answered using **commonsense knowledge** that most giraffes are brown. Additionally, these benchmarks may inadvertently capture an **imbalanced answer distribution**. For instance, in the MMStar benchmark [5] (which excludes questions solvable by blind LLMs like ChatGPT), "Yes" is three

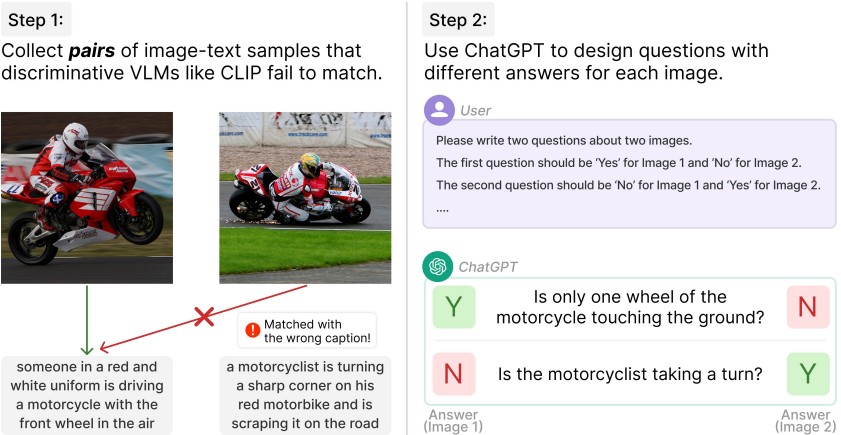

Figure 2: **Collecting NaturalBench.** We use a semi-automated procedure to collect NaturalBench from natural image-text corpora like Flickr30K [63]. First, we identify confounding pairs of image-text samples that fail discriminative VLMs like CLIP [65] and BLIP-2 [39], e.g., they wrongly match an image with another image's caption. Next, we prompt ChatGPT to design questions that yield different answers for each image, providing the original captions in the prompt. Section 3 details this procedure. We hire human annotators to filter out incorrect VQA samples, such as "*Is the motorcyclist wearing a red and white uniform?*", which has an identical answer of "Yes" for both images. Unlike previous adversarial benchmarks [20, 24, 40, 68], NaturalBench does not target any specific VQA models nor perturb the images or questions. Section 6 extends this simple procedure to diverse data sources (e.g., non-English) to highlight its potential for future dynamic evaluations [31] of VLMs.

times more likely than "No" to be the correct answer for yes-or-no questions. Section 4 shows that such spurious answer patterns allow one to achieve performance gains by finetuning a blind GPT-3.5 solely on QA data from these VQA benchmarks. To prevent blind solutions from scoring well, we introduce a *balanced* evaluation protocol: each test sample contains *two* images and *two* questions, with answers alternating between different questions or images. Consequently, blind solutions that choose the same answers regardless of the questions or images will not succeed under this protocol.

**NaturalBench.** We collect an initial benchmark with 5,800 yes-or-no and 1,800 multiple-choice VQA samples using public image-text datasets [59, 63], surpassing the scale of recent benchmarks like MMVP [73] and MMStar [5] (ranging from 300 to 1,500 samples). We hire a separate group of humans to evaluate themselves on NaturalBench tasks, achieving a high accuracy above 90%. We also evaluate over 50 open-source and closed-source VLMs. Popular models like LLaVA [48] and mPLUG-Owl [80] perform only marginally better than random chance, and even the best closed-source models such as GPT-4o and GPT-4Vision [60] lag significantly lag behind humans by more than 50%. This suggests that NaturalBench would serve as an effective testbed for driving future innovation in VLMs.

**What are the challenges?** We analyze why NaturalBench is difficult from two perspectives: (1) **Compositionality**: Solving NaturalBench requires diverse visio-linguistic skills [26, 32, 36, 71], such as attribute bindings, spatial/action/part relations, and advanced reasoning including comparison and logic. While most benchmarks assign only one skill tag per sample, we tag each sample with all applicable skills from a carefully defined taxonomy of 27 skills. Even the closed-source GPT-4o still struggles with certain skills such as spatial orientation and comparison, for example, "*Are the two people looking in the same direction?*". (2) **Biases**: NaturalBench reveals significant biases in VLMs, particularly their tendency to repeat the same answer across different images (or questions). Our analysis suggests debiasing is a promising way to ground VLMs and reduce hallucinations, with NaturalBench serving as a useful benchmark for bias mitigation [88].

**Dynamic evaluations.** To keep pace with model development and prevent data leakage [5, 78, 89], vision-language benchmarks must be continuously updated. Our benchmark curation method seamlessly adapts to dynamic evaluations [31, 58] by incorporating new data sources. We expand NaturalBench with over thousands of VQA samples constructed from two recent image-text datasets: (1) DOCCI [59] with detailed captions over 100 words, and (2) XM3600 [69] with non-English captions in Chinese and Hindi. Together, our first release of NaturalBench includes 10,000 samples,

presenting diverse challenges for next-generation VLMs. We hope our efforts will inspire further research into dynamic evaluations of VLMs.

## 2 Related Works

**Benchmarks for vision-language models.** Recent VLMs are commonly tested with popular VQA benchmarks such as MMStar [5], MMMU [83], MME [18], ScienceQA [53], AI2D [30], MM-Bench [50], MM-Vet [82], Seed-Bench [34], and MMVP [73]. These benchmarks evaluate complex visio-linguistic skills, such as fine-grained perception, reasoning and cognition, commonsense knowledge, and problem-solving across different fields. However, constructing them requires substantial human effort, including designing skills for evaluation, sourcing relevant images, and training annotators to create question-answer pairs [21, 40, 50, 68, 73, 82].

**Biases in vision-language benchmarks.** Despite careful construction, vision-language benchmarks are prone to spurious statistical patterns [21, 66] exploitable by "blind" shortcut solutions. For example, in the classic VQAv1 benchmark [2], questions starting with "Do you see a..." are answered "Yes" 87% of the time. Such language biases allow "blind" QA models to answer correctly without viewing images. While the community spent years addressing these issues, recent benchmarks designed for foundational VLMs still repeat these flaws [5, 45]. For example, image-text retrieval benchmarks like ARO [84] contain nonsensical negative captions that can be easily ruled out by caption likelihood [45, 74] or grammar correctness [26]. Recent VQA benchmarks like MMBench [50] are compromised by questions that can be solved using commonsense knowledge alone [5]. In this work, we show that even a blind ChatGPT can approach SOTA performance on these benchmarks, casting doubt on whether they truly assess *visio*-linguistic capabilities. As such, we design NaturalBench to avoid "blind" solutions by enforcing a balanced evaluation protocol [21, 45, 71].

**Adversarial samples for dynamic model evaluation.** Historically, machine learning models took decades to reach human performance on static benchmarks – for example, 15 years for MNIST [12] and 7 years for ImageNet [11]. However, modern foundation models [38, 60, 65] often make new benchmarks obsolete in just months or years [31]. In response, recent research advocates for dynamic (lifelong) benchmarking protocols [31, 43, 57, 70]. The most popular approach is to collect adversarial data samples through a human-and-model-in-the-loop procedure. For instance, Adversarial NLI [58] and Dynabench [31] engage human annotators to continuously craft hard samples that fail existing large language models. Similarly, adversarial VQA benchmarks [40, 68] ask humans to repeatedly write difficult QA pairs for an image until one fails a VQA model. Hendrycks et al. [24] train annotators to find web images that confuse pre-trained ImageNet classifiers. In contrast, our data collection method does not target any specific models and requires only single-step verification by human annotators, making it more efficient for dynamic benchmark curation.

## 3 Collecting NaturalBench

This section describes how we collect NaturalBench.

**Natural adversarial samples for VLMs.** For discriminative tasks like visual recognition, adversarial samples are *images* that models misclassify [20, 24]. For generative VLMs trained on tasks like VQA, we define their adversarial samples as *image-question pairs* that humans can easily answer but models cannot. Existing work often maliciously perturbs input images or prompts to compromise VLMs [16, 20, 27, 49, 55, 64]; instead, we challenge VLMs using natural image-question pairs.

**Challenges in designing VQA benchmarks.** Without careful curation, VQA benchmarks may be solved by blind QA models that ignore the images [5, 21]. First, Figure 3 shows that recent benchmarks often include questions solvable through **commonsense knowledge**. For example, MMBench [50] includes questions like, "*Is the African Elephant the smallest or largest land animal?*" which can be easily answered without seeing the image. Another question from MMMU [83] asks, "*Which artist belonging to the Bloomsbury group was Gertler in a relationship with?*" The correct answer is "Dora Carrington", as the other options like "Vanessa Bell" and "Leonora Carrington" can be ruled out with knowledge of art history. We also refer interested readers to the concurrent work [5] for further discussion on this issue. Another easily overlooked bias is **imbalanced answers**. For example, in the popular MME [18] benchmark, the question "*Does this artwork exist in the form of a painting?*" is answered "Yes" 97.5% of the time, while "*Does this artwork exist in the form*

*of furniture?*" is answered "No" 100% of the time. Even the concurrent MMStar [5] benchmark is not exempt from this issue, despite efforts to filter out samples from existing benchmarks that can be answered by blind language models. When MMStar asks about human emotions, "Sad" is three times more likely to be correct than "Happy". In MMStar's color-related questions, "White" and "Black" are up to ten times more likely to be correct than colors like "Purple". Section 4 shows that such spurious answer patterns can be exploited by finetuning a "blind" ChatGPT.

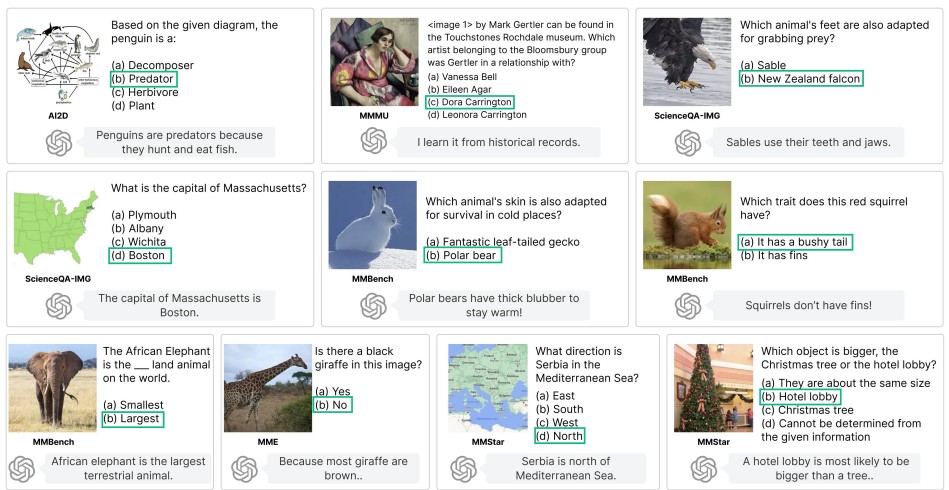

Figure 3: **Example questions in previous benchmarks solvable by commonsense knowledge.** We provide example questions from existing VQA benchmarks that can be addressed using commonsense knowledge. For these questions, a "blind" language model, such as ChatGPT (without vision input), can already answer them without looking at the image.

**Answer balancing (prior art).** To avoid blind solutions, a VQA benchmark must ensure that all candidate answers to a question are *equally* likely. However, balancing answer distribution is challenging. For example, GQA [28] performs post-hoc balancing by discarding over 90% of collected VQA samples. VQAv2 [21] uses a labor-intensive procedure to fix imbalances in VQAv1: for each (image, question, answer) triplet, MTurk annotators review up to 24 similar images (searched via nearest neighbor search) to find an alternative that leads to a different answer.

**Efficient construction of balanced VQA samples (ours).** We posit that foundation models [60, 65] can significantly reduce the human effort required to create balanced VQA benchmarks. Specifically, we automate the two most labor-intensive tasks: (1) finding pairs of similar images [21], and (2) generating corresponding questions and answers [40]. Given an image-text dataset like Flickr30K [63], our data curation pipeline (Figure 2) operates as follows. **Step 1:** We search for pairs of image-text samples that are incorrectly matched by discriminative VLMs like CLIP, meaning they erroneously match an image with another image's caption. **Step 2:** We use generative LLMs like ChatGPT to write questions that yield different answers for each image. We elaborate on this process below:

**Step 1: Collecting pairs of image-text samples.** We denote an image by $\mathbf{i}$ and a text caption by $\mathbf{t}$. VLMs like CLIP [65] compute a similarity score $S(\mathbf{i}, \mathbf{t}) \in \mathbb{R}$, with higher scores indicating greater similarity. For a pair of image-text samples $\{(\mathbf{i}_0, \mathbf{t}_0), (\mathbf{i}_1, \mathbf{t}_1)\}$, a correct match occurs when $S(\mathbf{i}_0, \mathbf{t}_0)$ and $S(\mathbf{i}_1, \mathbf{t}_1)$ are both greater than $S(\mathbf{i}_0, \mathbf{t}_1)$ and $S(\mathbf{i}_1, \mathbf{t}_0)$. Conversely, a mismatch occurs when:

$$[S(\mathbf{i}_0, \mathbf{t}_0) < S(\mathbf{i}_0, \mathbf{t}_1)] \quad \text{or} \quad [S(\mathbf{i}_0, \mathbf{t}_0) < S(\mathbf{i}_1, \mathbf{t}_0)] \quad \text{or} \quad [S(\mathbf{i}_1, \mathbf{t}_1) < S(\mathbf{i}_0, \mathbf{t}_1)] \quad \text{or} \quad [S(\mathbf{i}_1, \mathbf{t}_1) < S(\mathbf{i}_1, \mathbf{t}_0)] \quad (1)$$

Our Appendix shows that this adversarial procedure pairs visually and semantically similar images more efficiently than other methods (e.g., random pairing) for creating challenging VQA samples. Using Flickr30K [63], we identify all mismatches of both CLIP (`ViT-L-14-LAION400m`) [29] and BLIP-2 [39]. Since each sample can mismatch with multiple others, we randomly keep one mismatch per sample to collect about 2,000 unique pairs. We also hire annotators to discard around 800 pairs where a caption can describe both images. Our Appendix shows that these pairs already form an *image-text retrieval* benchmark in the same format as Winoground [71], challenging even the latest SigLIP [85] models trained with more parameters and data [67].

**Step 2: Generating questions and answers.** We ask ChatGPT to generate questions that yield different answers for two images. For example, given the caption pair $\mathbf{t}_0$ and $\mathbf{t}_1$, ChatGPT can generate questions answered "Yes" for one image and "No" for the other using the below instruction:

> I will present two captions for two images. Please help me generate two questions that highlight the differences between the captions. The first question should result in a 'Yes' answer for **Caption 1** and a 'No' for **Caption 2**. The second question should result in a 'No' answer for **Caption 1** and a 'Yes' for **Caption 2**.
> **Caption 1:** {$\mathbf{t}_0$}
> **Caption 2:** {$\mathbf{t}_1$}

Our Appendix shows how to modify this prompt for other question types (e.g., multiple-choice). For each generated VQA sample (a triplet of image, question, and answer), we engage two human annotators to select from the two candidate answers and "Unanswerable" [22]. We retain a sample only if both annotators agree on the correct answer. This step is crucial to ensure the quality of our benchmark. For samples that annotators fail or disagree on, we will resample new questions using ChatGPT up to three times. Using 1,200 Flickr30K image pairs, we manage to collect 2,600 yes-or-no and 1,000 multiple-choice VQA samples. Section 6 shows how NaturalBench can be easily expanded with new data, as we add 3,200 yes-or-no and 800 multiple-choice VQA samples collected from DOCCI. In the main paper, we present results on the combined dataset of 7,600 English VQA samples collected from Flickr30K [63] and DOCCI [59].

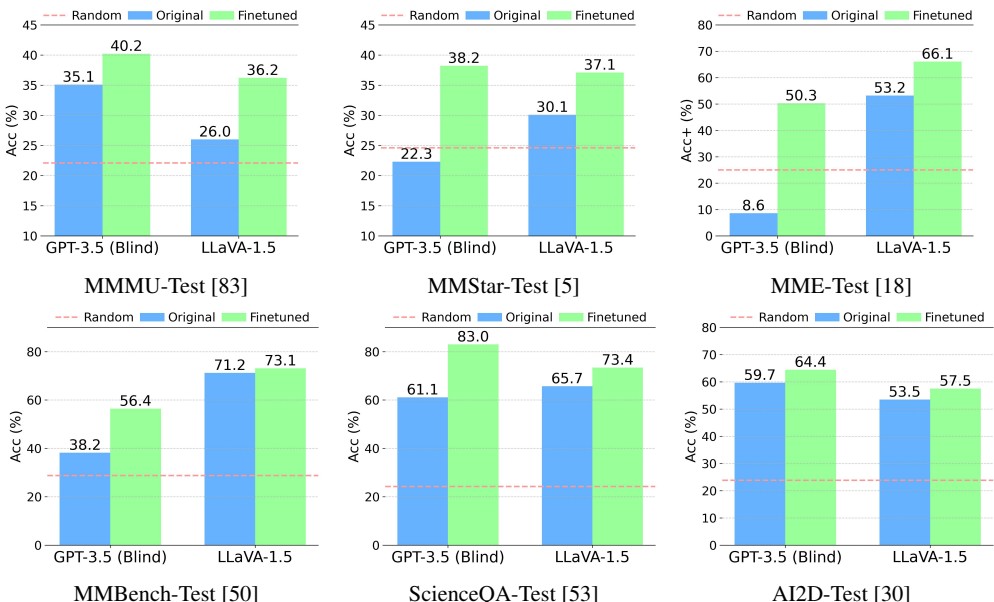

Figure 4: **Performance of GPT-3.5 vs. LLaVA-1.5 on previous VQA benchmarks.** We split each benchmark into equal-sized training and test sets, and report zero-shot (in blue) and finetuned (in green) results. Previous benchmarks show strong language biases, allowing blind GPT-3.5 to exploit spurious answer patterns (see Section 4) by finetuning on QA data without images. As a result, blind GPT-3.5 greatly surpasses random chance performance (see the red dotted line) and sometimes even matches the performance of LLaVA-1.5-7B finetuned using images. In contrast, Figure 5 shows that NaturalBench can effectively prevent blind solutions from exceeding chance.

## 4   Experimental Results

We present model results to contrast NaturalBench with previous benchmarks.

**NaturalBench is more robust against blind solutions.** Popular VQA benchmarks like MME [18] inadvertently *encourage* "blind" models that exploit language biases. We show this by using a random half of each benchmark for training and testing on the other half. We finetune a blind LLM (GPT-3.5) using auto (default) hyperparameters, while LLaVA-1.5 is finetuned with a learning rate of 2e-5 and a batch size of 16. Both models are trained for 10 epochs. Figure 4 shows that GPT-3.5 finetuned using only QA data (without images) significantly outperforms random chance and sometimes even

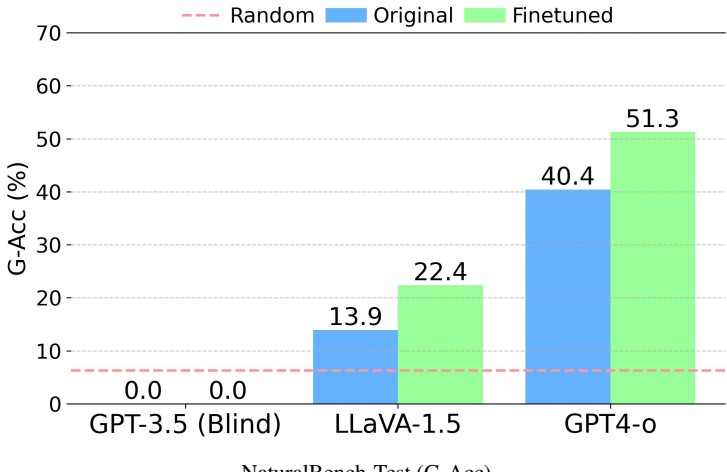

Figure 5: **Performance of GPT-3.5, LLaVA-1.5, and GPT-4o on NaturalBench.** We also split NaturalBench (the English subset) into equal-sized training and test sets, and report zero-shot (in blue) and finetuned (in green) results. We report group accuracy (**G-Acc**) (introduced in Section 4), which awards a point when all four (image, question) pairs are answered correctly. We highlight key results: (1) Blind GPT-3.5 fails to surpass random chance performance (red dotted line), regardless of finetuning. (2) LLaVA-1.5 improves by 9% by finetuning on NaturalBench's training images. (3) Even GPT-4o gains 10% G-Acc through vision finetuning on NaturalBench. These findings confirm that NaturalBench is a more vision-centric benchmark, and a potentially useful dataset for improving already advanced VLMs.

matches the performance of LLaVA-1.5 finetuned with images. In contrast, NaturalBench enforces a balanced answer distribution for each question and image. Figure 5 confirms that NaturalBench's design prevents blind GPT-3.5 from exceeding random chance performance, establishing it as a more *vision-centric* benchmark for reliable VLM evaluation. Additionally, vision finetuning of LLaVA-1.5 and GPT-4o [2] significantly boosts their performance over the original checkpoints, suggesting that NaturalBench is a potentially useful dataset for improving future VLMs. In this paper, we focus on model evaluation and leave data scaling for training to future work. We now proceed to evaluate more models on NaturalBench.

**Evaluation setup.** To better understand model performance, we introduce three additional metrics. We define the "**question accuracy**" (**Q-Acc**) metric to award a point only if a model correctly answers a question for *both* images. Similarly, the "**image accuracy**" (**I-Acc**) metric awards a point when a model correctly answers *both* questions for an image. Lastly, the "**group accuracy**" (**G-Acc**) metric awards one point when a model correctly answers all *four* (image, question) pairs in a test sample. These VQA metrics are analogous to Winoground's retrieval metrics [71] for paired (image, caption) samples. When reporting performance, we generate answers using each model's default decoding strategy and compare them to the ground-truth answers. Alternatively, Section 5 shows that models can be evaluated deterministically by comparing the likelihood of each candidate answer using VQAScore [44].

**NaturalBench challenges all state-of-the-art VLMs.** Table 1 shows that NaturalBench significantly challenges leading VLMs, with most models performing only 5% to 20% above random chance (in terms of G-Acc). Even models like InternLM-XC2-7B [15], despite being trained on Flickr30K images (but not our questions), perform only 20.2% better than chance. Closed-source models trained on proprietary datasets, such as GPT-4o and GPT-4v, still lag behind average human performance, as measured by a separate group of three human annotators. We also note that **Q-Acc** (correctly answering both questions for each image) is always lower than **I-Acc** (correctly answering both images for each question). This is primarily due to models choosing the same answer for a question regardless of the input image, which motivates our analysis in Section 5 on debiasing VLMs. Lastly, we observe that latest models like Qwen2-VL, Molmo, and Llama3.2 improve with larger language models. We now explore how NaturalBench identifies areas for future model improvement.

---

[2]Due to the high cost, we only finetune GPT-4o on NaturalBench, not other benchmarks. We use the 'auto' setting for vision finetuning of GPT-4o.

Table 1: **Performance on NaturalBench.** We report the performance of **53** leading VLMs on NaturalBench. All models significantly lag behind human performance, with the performance gap (in G-Acc) between humans and models highlighted in red. The latest models, such as BLIP-3 (XGen-MM), Cambrian-1, LLaVA-OneVision, Llama3.2-Vision, Molmo, and Qwen2-VL lag significantly behind humans by 55% to 70%. Even the best closed-source GPT-4o is still 52% behind humans.

| Model | Image Encoder | Language Model | NaturalBench Performance | | | | |
| --- | --- | --- | --- | --- | --- | --- | --- |
| | | | Acc | Q-Acc | I-Acc | G-Acc | $\Delta_{\text{Human}}$ |
| Human Performance | – | – | 97.5 | 94.6 | 95.0 | 92.1 | 0.0 |
| Random Chance | – | – | 50.0 | 25.0 | 25.0 | 6.3 | -85.8 |
| **Open-source Models** | | | | | | | |
| BLIP-2 [39] | EVA-G | FlanT5-3B | 56.2 | 14.0 | 17.1 | 2.1 | -89.9 |
| | | FlanT5-11B | 61.0 | 25.8 | 31.9 | 7.7 | -84.4 |
| InstructBLIP [9] | EVA-G | Vicuna-7B | 59.1 | 20.2 | 24.2 | 4.0 | -88.1 |
| | | Vicuna-13B | 62.8 | 29.0 | 34.8 | 9.2 | -82.9 |
| | | FlanT5-3B | 62.5 | 35.2 | 28.1 | 9.8 | -82.3 |
| | | FlanT5-11B | 64.5 | 32.8 | 39.1 | 12.7 | -79.4 |
| Otter [33] | CLIP-L-14 | MPT-7B | 57.4 | 20.9 | 24.9 | 3.8 | -88.3 |
| LlaMA-Adapter-v2.1 [19] | CLIP-L-14 | LaMA2-7B | 58.3 | 19.4 | 23.2 | 4.4 | -87.7 |
| CogVLM-Agent-VQA [25] | EVA2-E | Vicuna-7B | 64.9 | 31.1 | 34.7 | 10.3 | -81.8 |
| DeepSeek-VL-1.3B-Chat [51] | SigLIP-L & SAM-B | DeepSeek-LLM-1B | 66.5 | 35.4 | 39.4 | 11.5 | -80.6 |
| ShareGPT4V [4] | CLIP-L-14 | Vicuna-7B | 68.4 | 39.1 | 44.3 | 12.5 | -79.6 |
| | | Vicuna-13B | 69.3 | 40.5 | 44.3 | 14.9 | -77.2 |
| LLaVA-1.5 [47] | CLIP-L-14 | Vicuna-7B | 67.3 | 37.7 | 43.8 | 12.7 | -79.4 |
| | | Vicuna-13B | 68.9 | 39.6 | 44.6 | 14.8 | -77.3 |
| CogVLM-Chat [77] | EVA2-E | Vicuna-7B | 68.1 | 37.7 | 41.3 | 13.9 | -78.2 |
| InternLM-XC-V1 [87] | EVA-G | InternLM-7B | 68.7 | 40.3 | 46.9 | 15.5 | -76.6 |
| InternLM-XC-V2-1.8B [15] | CLIP-L-14 | InternLM2-1.8B | 70.5 | 43.3 | 46.9 | 16.6 | -75.5 |
| Qwen-VL-Chat [3] | CLIP-G-16 | Qwen-7B | 70.0 | 42.6 | 46.8 | 17.1 | -75.0 |
| Phi-3-Vision [1] | CLIP-L-14 | Phi-3-Mini | 70.4 | 43.4 | 48.7 | 17.2 | -74.9 |
| mPLUG-Owl2 [81] | CLIP-L-14 | Llama2-7B | 70.4 | 43.7 | 48.7 | 17.4 | -74.7 |
| Bunny [23] | SigLIP-SO | Phi-2-2.7B | 69.9 | 42.3 | 48.4 | 17.4 | -74.7 |
| mPLUG-Owl2.1 [81] | CLIP-L-14 | Qwen-7B | 70.1 | 42.5 | 47.1 | 17.9 | -74.2 |
| Monkey-10B-chat [41] | OpenCLIP-BigG | Qwen-7B | 71.1 | 43.9 | 48.3 | 18.2 | -73.9 |
| LLaVA-NeXT [48] | CLIP-L-14 | Vicuna-7B | 70.2 | 42.5 | 47.6 | 15.0 | -77.1 |
| | | Mistral-7B | 71.1 | 44.6 | 49.1 | 16.3 | -75.8 |
| | | Vicuna-13B | 72.2 | 45.9 | 49.9 | 19.2 | -72.9 |
| | | Nous-Hermes-2-Yi-34B | 73.5 | 48.2 | 50.9 | 22.7 | -69.4 |
| DeepSeek-VL-7B-Chat [51] | SigLIP-L & SAM-B | DeepSeek-LLM-7B | 71.7 | 46.0 | 50.1 | 19.3 | -72.8 |
| BLIP-3 (XGen-MM) [79] | CLIP-H-14 | Phi-3-Mini | 72.3 | 47.0 | 51.2 | 19.5 | -72.6 |
| InternVL-Chat-V1.1 [6] | InternViT-6B | Llama2-13B | 73.4 | 48.5 | 52.3 | 20.3 | -71.8 |
| InternVL-Chat-V1.5 [7] | InternViT-6B | InternLM2-Chat-20B | 75.3 | 52.3 | 55.9 | 23.1 | -69.0 |
| InternVL-Chat-V1.2-Plus [6] | InternViT-6B | Nous-Hermes-2-Yi-34B | 75.5 | 52.7 | 56.2 | 23.4 | -68.7 |
| InternVL2-8B [8] | InternViT-300M | InternLM2.5-7B-Chat | 74.0 | 50.4 | 54.4 | 23.5 | -68.6 |
| Cambrian-1 [72] | SigLIP-S-14 & CLIP-L-14 | Llama-3-8B | 71.5 | 44.6 | 47.9 | 19.4 | -72.7 |
| | DINOv2-g & CLIP-ConvNeXT-XXL | Vicuna-13B | 75.4 | 52.6 | 55.7 | 25.5 | -66.6 |
| | | Nous-Hermes-2-Yi-34B | 76.3 | 53.9 | 57.2 | 26.6 | -65.5 |
| InternLM-XC2-4KHD-7B [13] | CLIP-L-14 | InternLM2-7B | 75.5 | 53.1 | 56.1 | 25.9 | -66.2 |
| InternLM-XC2-7B [15] | CLIP-L-14 | InternLM2-7B | 76.0 | 53.9 | 56.7 | 26.5 | -65.6 |
| InternVL-Chat-V1.2 [6] | InternViT-6B | Nous-Hermes-2-Yi-34B | 75.6 | 52.9 | 56.4 | 26.6 | -65.5 |
| InternVL2-26B [8] | InternViT-6B | InternLM2-Chat-20B | 76.9 | 55.4 | 58.4 | 27.7 | -64.4 |
| LLaVA-OneVision [37] | SigLIP-S-14 | Qwen2-0.5B | 68.7 | 39.8 | 46.2 | 15.6 | -76.5 |
| | | Qwen2-7B | 77.2 | 56.1 | 58.8 | 28.8 | -63.3 |
| Llama3.2-Vision [17] | ViT-H-14 | Llama-3.1-8B | 75.1 | 51.7 | 55.6 | 26.8 | -65.3 |
| | | Llama-3.1-70B | 77.0 | 55.1 | 57.5 | 29.1 | -63.0 |
| Molmo [10] | CLIP-L-14 | OLMoE-1B | 68.3 | 38.2 | 42.6 | 14.7 | -77.4 |
| | | OLMo-7B | 72.8 | 46.8 | 50.3 | 20.7 | -71.5 |
| | | Qwen2-7B | 75.3 | 52.0 | 55.8 | 26.7 | -65.4 |
| | | Qwen2-72B | 76.4 | 53.9 | 57.0 | 29.3 | -62.8 |
| Qwen2-VL [76] | CLIP-L-14 | Qwen2-1.5B | 74.1 | 50.8 | 54.4 | 23.4 | -68.7 |
| | | Qwen2-7B | 76.7 | 55.5 | 58.5 | 29.1 | -63.0 |
| | | Qwen2-72B | 79.9 | 61.3 | 64.0 | 36.9 | -55.2 |
| **Closed-source Models** | | | | | | | |
| GPT-4Vision | | GPT-4 | 75.0 | 52.5 | 56.1 | 26.2 | -65.9 |
| GPT-4o | | GPT-4 | **81.6** | **64.4** | **66.4** | **39.6** | **-52.5** |

# 5   Why is NaturalBench Challenging?

We analyze why NaturalBench is challenging from (1) **compositionality** and (2) **biases**.

**NaturalBench assesses visio-linguistic compositional reasoning.** Solving a NaturalBench sample often requires a combination of skills, including object recognition, attribute binding, relation understanding, and advanced reasoning such as logic, comparison, differentiation (instance discrimination), counting, and world knowledge. For fine-grained evaluation, we manually tag each (image, question) pair with *all* associated skills, unlike prior benchmarks that oversimplify by assigning a single skill

tag per sample. Figure 6 showcases the skill taxonomy with 8 types of objects, 8 types of attributes, 3 types of relations (with spatial relation further divided into 4 subtypes [46]), and 5 types of reasoning [35]. This taxonomy is more compositional than previous benchmarks such as MMVP [73], which assigns each sample a single tag from 8 skill types. The Appendix provides detailed skill definitions, examples, and model performance for each skill. Several conclusions are noted: (1) Certain skills are generally harder than others; for example, *abstract* attributes (e.g., helpful) are harder than *color* or *size* attributes. *Orientation* (e.g., facing, towards) is harder than other spatial relations like *proximity* (e.g., near, far) or *projectivity* (e.g., to the right, on top of). (2) Even the strongest GPT-4o are challenged by advanced reasoning skills such as *comparison* (e.g., more than, the same as, happier).

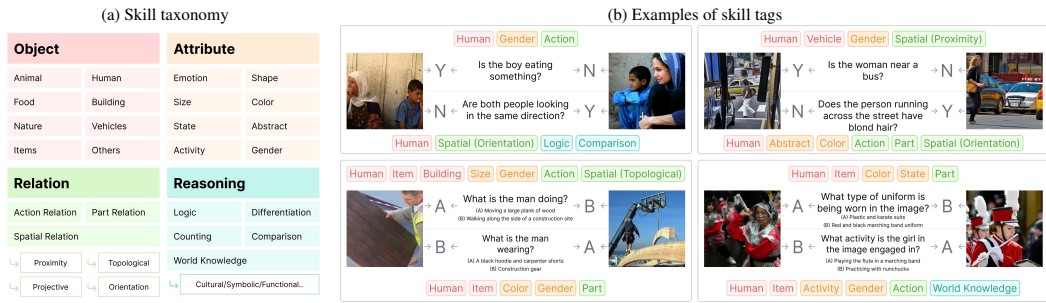

Figure 6: **Skill taxonomy and tagging. Figure (a)** provides an overview of the compositional reasoning skills evaluated by NaturalBench. Skill definitions and model performance per skill are presented in our Appendix. **Figure (b)** shows NaturalBench samples with their associated skill tags. Unlike prior benchmarks that assign a single tag per sample, NaturalBench tags each sample with all applicable skills for a fine-grained analysis.

**NaturalBench exposes significant model biases.** We find that most VLMs underperform on NaturalBench due to biases towards certain answers, such as "Yes" for yes-or-no questions and "B" for multiple-choice questions. We posit that mitigating these biases can improve model performance. To show this, we adopt a scoring-based evaluation strategy using the generative likelihood of each candidate answer (VQAScore [44]) to determine the model's predicted answer. Specifically, given a question $q$, an image $i$, and two candidate answers $a_0$ and $a_1$, we evaluate:

$$P(a_0|q,i) - P(a_1|q,i) > \tau \tag{2}$$

where $\tau$ is a threshold (default is 0). If this condition (Eq. 2) is met, the model predicts $a_0$; otherwise, it predicts $a_1$. The Appendix shows that deterministic evaluation yields results largely consistent with stochastic decoding while being more reproducible. Crucially, this formulation allows us to adjust $\tau \in [-1, 1]$ for each NaturalBench sample (four image-question pairs) to avoid repeating the same answers across images (or questions). Recall that **Q-Acc** awards a point when the model correctly answers both images for a question. We can now calculate a **debiased Q-Acc** by adjusting $\tau$ so that the model predicts different answers for each image. Similarly, a **debiased I-Acc** is calculated by adjusting $\tau$ to ensure different predicted answers for each question (of the same image). For **debiased G-Acc**, we tune $\tau$ to make the model predict $a_0$ for two of the four image-question pairs and $a_1$ for the other two pairs. Table 2 shows that these metrics significantly outperform the original ones by 35% to 40%, indicating that proper debiasing of the model can lead to notable performance gains. Importantly, our debiased metrics reflect the ability of a VLM to correctly *rank* the set of eight image-question-answer triples, such that the correct combinations are more probable than incorrect ones. However, this protocol violates the original task of answering a single image-question pair. This motivates us to study alternate debiasing techniques [88] in the Appendix. We believe NaturalBench could be a promising testbed for techniques to ground VLMs and reduce biased responses (hallucinations).

## 6 Extending to Dynamic Evaluation

We now show our benchmark curation method can facilitate dynamic evaluation [31, 70].

**Expanding NaturalBench.** Since benchmarks often leak into foundation models' training data, it is crucial to update benchmarks using new data sources. Our benchmark curation method can easily adapt to new image-text datasets. We expand NaturalBench by incorporating two recently proposed

Table 2: **Debiased performance on NaturalBench.** Many models underperform on NaturalBench due to biases towards certain answers like "Yes" and "B". To illustrate this, we compute a **debiased Q-Acc** by adjusting the prediction threshold (as described in Section 5) to ensure the model predict different answers for the two images of the same question. Similarly, **debiased I-Acc** ensures different predicted answers for the two questions of the same image. For **debiased G-Acc**, we tune the threshold so that the model predicts one answer for two (out of four) image-question pairs, and a different answer for the other two pairs. The substantial performance gains of these metrics suggest that proper debiasing can greatly improve performance. Our Appendix evaluates existing debiasing techniques that do not require prior knowledge of image-question pairings.

| Model | Image Encoder | Language Model | Q-Acc | | I-Acc | | G-Acc | |
|---|---|---|---|---|---|---|---|---|
| | | | Original | Debiased | Original | Debiased | Original | Debiased |
| LLaVA-1.5 | CLIP-L-14 | Vicuna-13B | 38.6 | **86.2** | 43.5 | **78.6** | 14.4 | **49.7** |
| DeepSeek-VL-7B-Chat | SigLIP-L | SAM-B | 45.8 | **86.6** | 49.9 | **81.8** | 19.4 | **54.8** |
| BLIP-3 (XGen-MM) | CLIP-H-14 | Phi-3-Mini | 46.8 | **88.6** | 51.1 | **81.9** | 19.5 | **55.3** |
| InternVL-Chat-V1.5 | InternViT-6B | InternLM2-Chat-20B | 52.6 | **92.3** | 56.0 | **86.1** | 24.3 | **66.0** |
| InternVL-Chat-V1.2 | InternViT-6B | Nous-Hermes-2-Yi-34B | 52.6 | **91.6** | 56.0 | **86.0** | 26.2 | **65.8** |
| InternVL2-26B | InternViT-6B | InternLM2-Chat-20B | 55.7 | **92.2** | 58.5 | **87.2** | 28.2 | **67.7** |
| LLaVA-OneVision | SigLIP-S-14 | Qwen2-7B | 55.4 | **92.1** | 58.2 | **87.2** | 28.6 | **67.8** |
| GPT-4o | - | GPT-4 | 65.0 | **94.0** | 67.0 | **90.5** | 40.5 | **75.6** |

Table 3: **NaturalBench statistics.** We report model performance on each dataset in the Appendix.

| Benchmark Statistics | | | | Collection Details | | |
|---|---|---|---|---|---|---|
| Source | Question Type | Language | # VQA Samples | # VLMs Used | # Mismatched Pairs | # Verified Pairs |
| **NaturalBench** | | | | | | |
| Flickr30K [63] | Yes-or-No | English | 2,600 | CLIP-L, BLIP-2, GPT-4 | 2,000 | 1,200 |
| Flickr30K [63] | Multiple-Choice | English | 1,000 | CLIP-L, BLIP-2, GPT-4 | 2,000 | 1,200 |
| DOCCI [59] | Yes-or-No | English | 3,200 | LongCLIP, GPT-4 | 3,300 | 1,000 |
| DOCCI [59] | Multiple-Choice | English | 800 | LongCLIP, GPT-4 | 3,300 | 1,000 |
| *All* | *Yes-or-No, Multiple-Choice* | *English* | *7,600* | - | - | - |
| **NaturalBench (Multi-lingual)** | | | | | | |
| XM3600 [69] | Yes-or-No | Chinese | 1,200 | NLLB-CLIP, GPT-4 | 2,400 | 400 |
| XM3600 [69] | Yes-or-No | Hindi | 1,200 | NLLB-CLIP, GPT-4 | 2,400 | 400 |
| *All* | *Yes-or-No* | *Chinese, Hindi* | *2,400* | - | - | - |

datasets: (1) DOCCI [59] with fine-grained captions over 100 words, and (2) XM3600 [69] with captions in Chinese and Hindi. Specifically, we use the latest longCLIP [86] for processing long text sequences and NLLB-CLIP [75] for non-English captions. Our Appendix details the collection process, model performance, and prompts used to collect VQA samples with ChatGPT. Table 3 shows all benchmarks we have collected. We hope our efforts will inspire future work in studying dynamic evaluations of VLMs.

# 7 Conclusion

**Limitations.** Our collected samples may inherit biases from web-scraped datasets and foundation models [56, 62], making human verification crucial. While this work focuses on model performance for individual skill tags, future work may analyze performance using combinations of skills.

**Summary.** We introduce **NaturalBench** to evaluate vision-language models on their *natural adversarial samples* – samples that challenge models significantly more than humans. Unlike previous benchmarks where "blind" models could succeed without the images, NaturalBench better reflects VLMs' genuine progress by penalizing solutions that ignore images. Furthermore, NaturalBench offers comprehensive skill tags to assess compositional reasoning abilities and highlights model biases in VLMs. Lastly, we show that our semi-automated method for benchmark curation can adapt to new data sources, facilitating future dynamic evaluations of VLMs.

# 8 Acknowledgement

We thank Pengchuan Zhang, Emily Li, and Anoushka Shrivastava for their invaluable discussions during the development of this work. We thank Tiffany Ling for her contribution to the visual design.

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

# NaturalBench: Evaluating Vision-Language Models on Natural Adversarial Samples

## Supplementary Material

### *Outline*

This document supplements the main paper with detailed results. Below is the outline:

- **Section A** details the collection process of NaturalBench.
- **Section B** details VQA and image-text retrieval performance on NaturalBench.
- **Section C** provides skill definitions and analyzes model performance by skills.
- **Section D** reports other debiasing techniques on NaturalBench.

## A    Collection Details

We provide further details on the collection pipeline.

**Step 1: Collecting pairs of image-text samples.** We collect pairs of image-text samples by finding mismatches of discriminative VLMs like CLIP. Recall that VLMs like CLIP [65] compute a similarity score $S(\mathbf{i}, \mathbf{t}) \in \mathbb{R}$, with higher scores indicating greater similarity between the image $\mathbf{i}$ and text caption $\mathbf{t}$. For a pair of image-text samples $\{(\mathbf{i}_0, \mathbf{t}_0), (\mathbf{i}_1, \mathbf{t}_1)\}$, a mismatch occurs when:

$$[S(\mathbf{i}_0, \mathbf{t}_0) < S(\mathbf{i}_0, \mathbf{t}_1)] \quad \text{or} \quad [S(\mathbf{i}_0, \mathbf{t}_0) < S(\mathbf{i}_1, \mathbf{t}_0)] \quad \text{or} \quad [S(\mathbf{i}_1, \mathbf{t}_1) < S(\mathbf{i}_0, \mathbf{t}_1)] \quad \text{or} \quad [S(\mathbf{i}_1, \mathbf{t}_1) < S(\mathbf{i}_1, \mathbf{t}_0)] \tag{3}$$

Importantly, this adversarial procedure efficiently pairs similar image-text samples for two purposes. First, these image-text pairs already form an image-text retrieval task that can be evaluated using Winoground's [71] evaluation protocols (after removing pairs where one caption can describe both images). We term this benchmark **NaturalBench-Retrieval** and report the performance of CLIP and SigLIP in Table 7. Next, by considering both images and captions, we can pair samples that are semantically similar but not necessarily visually similar. This contrasts with MMVP [73] which only pairs visually similar images close in DINO's feature space.

**Implementation of step 1.** For Flickr30K [63], we retrieve pairs mismatched by both OpenCLIP (LAION400M-ViT-L14) [29] and BLIP-2 (ViT-L) [39]. For DOCCI [59], we use both longCLIP-B and longCLIP-L. However, since DOCCI's captions are still too long to process, we use ChatGPT to shorten them to below 230 characters per caption. We believe future advances in long-context CLIP will streamline this process. Lastly, for XM3600, we use NLLB-CLIP [75] to process the Chinese and Hindi captions.

**Step 2: Generating questions and answers.** We use ChatGPT to generate questions that yield different answers for two images using their textual captions. We now show the actual prompts we send to ChatGPT.

**Default instruction for GPT-4.** In practice, we use the below prompt to ask GPT-4 to directly output a JSON dictionary for easier processing:

> I will present two captions for two images. Please help me generate two questions that highlight the differences between the captions. The first question should result in a 'Yes' answer for **Caption 1** and a 'No' for **Caption 2**. The second question should result in a 'No' answer for **Caption 1** and a 'Yes' for **Caption 2**.
> **Caption 1:** {$\mathbf{t}_0$}
> **Caption 2:** {$\mathbf{t}_1$}
> Please response in JSON format with question indices as the keys, starting from 0 and question-answer pairs {{"Question":...,"Caption1 Answer":...,"Caption2 Answer":...}} as the values.

**Instructions for generating Chinese and Hindi QA pairs.** We can simply ask GPT-4 to generate questions and answers in Chinese and Hindi:

I will present two captions for two images. Please help me generate two questions in Chinese / Hindi that highlight the differences between the captions. The first question should result in a 'Yes' answer for **Caption 1** and a 'No' for **Caption 2**. The second question should result in a 'No' answer for **Caption 1** and a 'Yes' for **Caption 2**.
**Caption 1: {$t_0$}**
**Caption 2: {$t_1$}**
Please response in JSON format with question indices as the keys, starting from 0 and question-answer pairs {{"Question":...,"Caption1 Answer":...,"Caption2 Answer":...}} as the values.

**Instructions for generating multiple-choice QA pairs.** We ask ChatGPT to generate multiple-choice questions using the below prompt:

I will present two captions for two images. Please help me generate two multiple-choice questions that highlight the differences between the captions. Each question should have options A and B. For the first question, option A corresponds to **Caption 1** and option B corresponds to **Caption 2**. For the second question, option A corresponds to **Caption 2** and option B corresponds to **Caption 1**.
**Caption 1: {$t_0$}**
**Caption 2: {$t_1$}**
Please response in JSON format with question indices as the keys, starting from 0 and question-answer pairs {{"Question":...,"Caption1 Answer":...,"Caption2 Answer":...}} as the values.

We engage two human annotators to select from the two candidate answers and "Unanswerable" [22] for all generated QA pairs, retaining a sample only if both annotators agree on the correct answer. In total, we spend around 500 annotator hours to collect all samples at 14 dollars per hour. For the Chinese and Hindi subsets, the authors (who are native speakers of these languages) manually examine all the questions.

**Additional examples.** Figure 7 provides additional examples of NaturalBench.

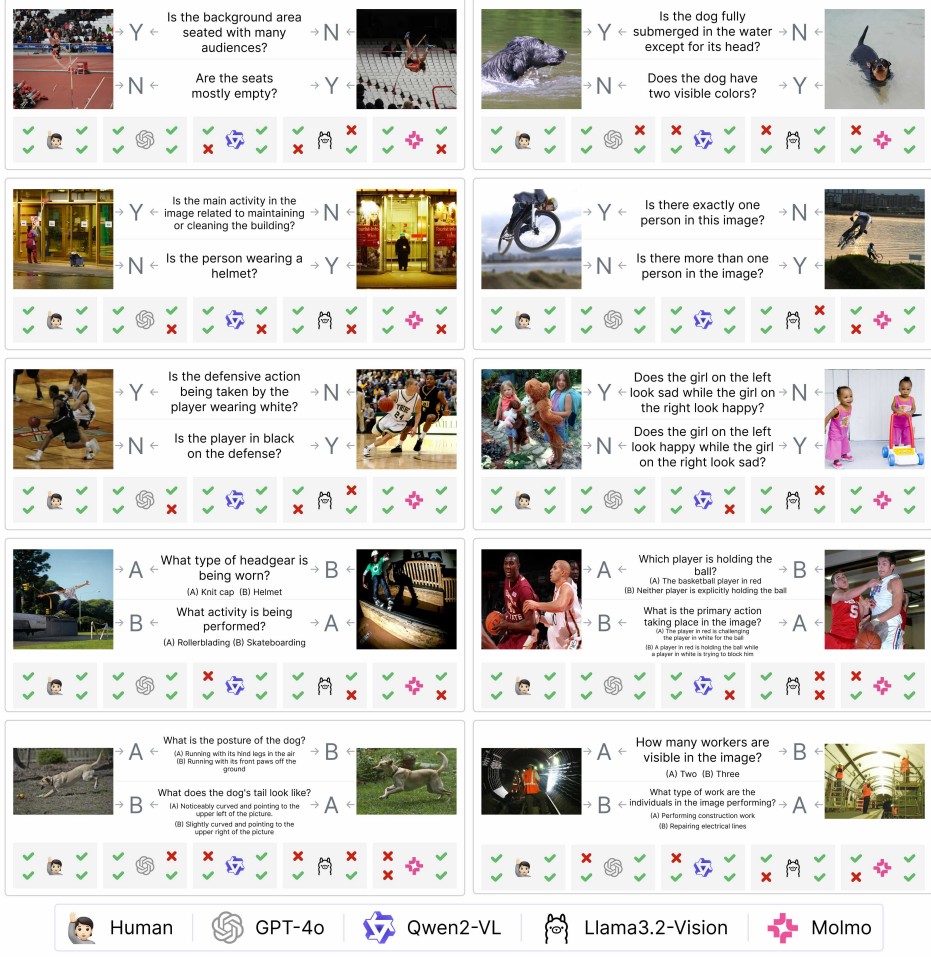

Figure 7: **More NaturalBench examples.**

# B   NaturalBench Performance

We report model performance on different subsets of NaturalBench.

**Performance on different subsets.** Table 4 reports **G-Acc** on subsets of NaturalBench.

Table 4: **Performance on different subsets of NaturalBench.** We report the **G-Acc** performance of 53 leading VLMs on subsets of NaturalBench.

| Model | Image Encoder | Language Model | NaturalBench Performance | | | | |
|---|---|---|---|---|---|---|---|
| | | | Flickr-YN | Flickr-MCQ | DOCCI-YN | DOCCI-MCQ | Overall |
| Human Performance | – | – | 91.5 | 92.0 | 92.2 | 93.9 | 92.1 |
| Random Chance | – | – | 6.3 | 6.3 | 6.3 | 6.3 | 6.3 |
| **Open-source Models** | | | | | | | |
| BLIP-2 [39] | EVA-G | FlanT5-3B | 2.7 | 0.8 | 2.3 | 0.5 | 2.1 |
| | | FlanT5-11B | 6.1 | 3.2 | 12.1 | 1.0 | 7.7 |
| InstructBLIP [9] | EVA-G | Vicuna-7B | 2.9 | 0.4 | 6.8 | 0.5 | 4.0 |
| | | Vicuna-13B | 7.0 | 0.4 | 14.8 | 5.0 | 9.2 |
| | | FlanT5-3B | 9.6 | 1.2 | 15.1 | 0.5 | 9.8 |
| | | FlanT5-11B | 12.6 | 2.8 | 18.6 | 2.5 | 12.7 |
| Otter [33] | CLIP-L-14 | MPT-7B | 3.7 | 4.0 | 4.5 | 1.5 | 3.8 |
| LlaMA-Adapter-v2.1 [19] | CLIP-L-14 | LlaMA2-7B | 4.2 | 1.2 | 6.6 | 0.5 | 4.4 |
| CogVLM-Agent-VQA [25] | EVA2-E | Vicuna-7B | 12.2 | 2.8 | 13.6 | 0.5 | 10.3 |
| DeepSeek-VL-1.3B-Chat [51] | SigLIP-L & SAM-B | DeepSeek-LLM-1B | 7.8 | 3.6 | 15.5 | 17.0 | 11.5 |
| LLaVA-1.5 [47] | CLIP-L-14 | Vicuna-7B | 9.1 | 14.8 | 14.1 | 16.5 | 12.7 |
| | | Vicuna-13B | 9.1 | 21.2 | 15.1 | 24.0 | 14.8 |
| ShareGPT4V | CLIP-L-14 | Vicuna-7B | 10.0 | 13.2 | 12.9 | 18.5 | 12.5 |
| | | Vicuna-13B | 9.5 | 19.6 | 15.6 | 23.5 | 14.9 |
| CogVLM-Chat [77] | EVA2-E | Vicuna-7B | 14.6 | 15.6 | 14.5 | 7.5 | 13.9 |
| InternLM-XC-V1 [87] | EVA-G | InternLM-7B | 11.5 | 16.8 | 15.2 | 28.0 | 15.5 |
| InternLM-XC-V2-1.8B [15] | CLIP-L-14 | InternLM2-1.8B | 12.0 | 25.6 | 15.1 | 26.5 | 16.6 |
| Qwen-VL-Chat [3] | CLIP-G-16 | Qwen-7B | 16.0 | 16.8 | 16.9 | 21.5 | 17.1 |
| Phi-3-Vision [1] | CLIP-L-14 | Phi-3-Mini | 15.4 | 17.6 | 15.3 | 30.0 | 17.2 |
| mPLUG-Owl2 [81] | CLIP-L-14 | LlaMA2-7B | 14.0 | 20.0 | 17.3 | 25.5 | 17.4 |
| Bunny [23] | SigLIP-SO | Phi-2-2.7B | 12.0 | 16.8 | 18.9 | 30.0 | 17.4 |
| mPLUG-Owl2.1 [81] | CLIP-L-14 | Qwen-7B | 12.3 | 20.0 | 17.4 | 36.0 | 17.9 |
| Monkey-10B-chat [41] | OpenCLIP-BigG | Qwen-7B | 17.1 | 12.0 | 19.5 | 24.0 | 18.2 |
| LLaVA-NeXT [48] | CLIP-L-14 | Vicuna-7B | 12.5 | 17.6 | 14.5 | 22.0 | 15.0 |
| | | Mistral-7B | 13.7 | 21.6 | 14.6 | 24.5 | 16.3 |
| | | Vicuna-13B | 15.7 | 22.8 | 19.0 | 26.5 | 19.2 |
| | | Nous-Hermes-2-Yi-34B | 16.2 | 32.0 | 20.8 | 40.0 | 22.7 |
| DeepSeek-VL-7B-Chat [51] | SigLIP-L & SAM-B | DeepSeek-LLM-7B | 13.8 | 18.8 | 21.6 | 28.5 | 19.3 |
| BLIP-3 (XGen-MM) [79] | CLIP-H-14 | Phi-3-Mini | 13.7 | 19.2 | 21.6 | 30.5 | 19.5 |
| InternVL-Chat-V1.1 [6] | InternViT-6B | LlaMA2-13B | 19.7 | 21.6 | 16.5 | 36.0 | 20.3 |
| InternVL-Chat-V1.5 [7] | InternViT-6B | InternLM2-Chat-20B | 22.5 | 32.8 | 17.4 | 35.5 | 23.1 |
| InternVL-Chat-V1.2-Plus [6] | InternViT-6B | Nous-Hermes-2-Yi-34B | 26.5 | 31.2 | 17.0 | 29.5 | 23.4 |
| InternVL2-8B [8] | InternViT-300M | InternLM2.5-7B-Chat | 20.7 | 34.8 | 19.5 | 35.0 | 23.5 |
| Cambrian-1 [72] | SigLIP-S-14 & CLIP-L-14 | Llama-3-8B | 16.2 | 15.6 | 24.6 | 14.0 | 19.4 |
| | DINOv2-g & | Vicuna-13B | 19.6 | 30.8 | 26.1 | 35.5 | 25.5 |
| | CLIP-ConvNeXT-XXL | Nous-Hermes-2-Yi-34B | 23.8 | 35.2 | 23.7 | 36.5 | 26.6 |
| InternLM-XC2-4KHD-7B [13] | CLIP-L-14 | InternLM2-7B | 22.8 | 33.2 | 24.3 | 34.0 | 25.9 |
| InternLM-XC2-7B [15] | CLIP-L-14 | InternLM2-7B | 25.4 | 38.4 | 21.8 | 34.0 | 26.5 |
| InternVL-Chat-V1.2 [6] | InternViT-6B | Nous-Hermes-2-Yi-34B | 21.8 | 34.4 | 24.5 | 41.0 | 26.6 |
| InternVL2-26B [8] | InternViT-6B | InternLM2-Chat-20B | 26.6 | **40.4** | 22.1 | 37.5 | 27.7 |
| LLaVA-OneVision [37] | SigLIP-S-14 | Qwen2-0.5B | 12.0 | 14.4 | 18.6 | 17.5 | 15.6 |
| | | Qwen2-7B | 27.0 | 32.8 | 26.0 | 41.5 | 28.8 |
| Llama3.2-Vision [17] | ViT-H-14 | Llama-3.1-8B | 16.2 | 29.2 | 30.0 | 45.5 | 26.8 |
| | | Llama-3.1-70B | 23.7 | 37.2 | 24.8 | **53.5** | 29.1 |
| Molmo [10] | CLIP-L-14 | OLMoE-1B-7B | 10.8 | 15.2 | 14.3 | 29.0 | 14.7 |
| | | OLMo-7B | 14.6 | 24.0 | 20.6 | 37.0 | 20.7 |
| | | Qwen2-7B | 20.6 | 31.2 | 25.8 | 44.5 | 26.7 |
| | | Qwen2-72B | 23.5 | 38.4 | 25.3 | 52.5 | 29.3 |
| Qwen2-VL [76] | CLIP-L-14 | Qwen2-1.5B | 17.4 | 22.8 | 25.6 | 35.0 | 23.4 |
| | | Qwen2-7B | 18.8 | 28.4 | 32.9 | 48.5 | 29.1 |
| | | Qwen2-72B | 28.2 | 36.0 | **40.5** | 52.0 | 36.9 |
| **Closed-source Models** | | | | | | | |
| GPT-4Vision | – | GPT-4 | 22.8 | 25.2 | 26.9 | 36.0 | 26.2 |
| GPT-4o | – | GPT-4 | **37.5** | **40.4** | 39.0 | 48.0 | **39.6** |

**Performance on NaturalBench-Hindi and NaturalBench-Chinese.** Table 5 reports the performance on the multilingual subsets of NaturalBench, evaluating only the models that claim to have multilingual capabilities. We also report the performance of these datasets after using ChatGPT to translate the questions and answers into English. This shows that most models are still better at solving English VQA tasks.

**Ablation on samples generated by different methods.** Table 6 reports **G-Acc** on two types of generated VQA samples: (1) **Flickr-Adversarial**, generated by sending caption pairs to GPT-4, (2)

Table 5: **Performance on NaturalBench-Chinese and NaturalBench-Hindi.** We report **G-Acc** for each dataset, evaluating only models with claimed multilingual capabilities. For both datasets, we also provide G-Acc after translating the original Chinese or Hindi questions into English. This simple translation often boosts performance, except for top models like InternVL-Chat-V1.2-Plus and GPT-4o, which seem extensively trained in Chinese. NaturalBench-Hindi remains particularly challenging for open-source models.

| Model | NaturalBench-Chinese | | NaturalBench-Hindi | |
|---|---|---|---|---|
| | Chinese | English | Hindi | English |
| Random Chance | 6.3 | 6.3 | 6.3 | 6.3 |
| **Open-source Models** | | | | |
| DeepSeek-VL-7B-Chat | 10.9 | **28.4** | 0.6 | **29.0** |
| InternVL-Chat-V1.2-Plus | **34.6** | 33.4 | 11.5 | **36.2** |
| InternLM-XC2-7B | 32.5 | **34.6** | 15.9 | **35.6** |
| **Closed-source Models** | | | | |
| GPT-4o | **41.2** | 38.7 | 40.3 | **40.9** |

Table 6: **Ablation on different collection methods.** We report **G-Acc** on datasets generated by different collection methods from Flickr30K. Our adversarial procedure results in a much more challenging dataset. Note that Flickr-Adversarial is the combination of Flickr-YN and Flickr-MCQ.

| Model | Model Performance (G-Acc) | |
|---|---|---|
| | Flickr-Adversarial | Flickr-Random |
| Random Chance | 6.3 | 6.3 |
| **Open-source Models** | | |
| DeepSeek-VL-7B-Chat | 15.2 | 80.7 |
| BLIP-3(XGen-MM) | 15.2 | 69.0 |
| LLaVA-NeXT (Mistral-7B) | 15.9 | 86.0 |
| Phi-3-Vision | 16.0 | 75.0 |
| InternVL-Chat-V1.2-Plus | 27.8 | 83.0 |
| InternLM-XC2-7B | 29.0 | 84.5 |
| **Closed-source Models** | | |
| GPT-4o | 38.3 | 72.5 |

**Flickr-Random**, generated by sending caption pairs of *randomly matched* image-text samples to GPT-4. The results confirm that it is crucial to use discriminative VLMs to first search for confounding pairs of image-text samples.

**Performance on NaturalBench-Retrieval.** Table 7 reports model performance on NaturalBench-Retrieval. We only use Flickr image-text samples to construct this benchmark. We adopt the evaluation metrics proposed by Winoground [71].

Table 7: **Image-text retrieval performance on NaturalBench-Retrieval.** We evaluate CLIP and SigLIP models on the human-verified 1,200 paired (image, text) samples from NaturalBench-Flickr. We follow Winoground [71] to report text score, image score, and group score, with higher numbers indicating better performance for all metrics. We exclude the CLIP (LAION400M-ViT-L14) model used to collect these adversarial pairs. Overall, NaturalBench-Retrieval poses a significant challenge to leading discriminative models.

| Method | Source | Model | Data Size | Model Size (M) | Retrieval Performance | | |
|---|---|---|---|---|---|---|---|
| | | | | | Group | Image | Text |
| Random | – | – | – | – | 16.67 | 25.00 | 25.00 |
| CLIP [65] | OpenAI | RN50 | 400M | 102 | 12.22 | 32.60 | 36.76 |
| | | RN101 | | 120 | 13.61 | 35.04 | 33.33 |
| | | ViT-B-32 | | 151 | 15.89 | 36.43 | 36.92 |
| | | RN50x4 | | 178 | 14.75 | 37.49 | 36.27 |
| | | RN50x16 | | 291 | 24.61 | 44.01 | 43.93 |
| | | ViT-L-14 | | 428 | 23.15 | 44.99 | 41.81 |
| | | RN50x64 | | 623 | 26.24 | 46.21 | 47.35 |
| | LAION | roberta-ViT-B-32 | 2B | 212 | 16.22 | 39.36 | 38.79 |
| | | ViT-H-14 | | 986 | 24.04 | 49.31 | 48.82 |
| | | ViT-g-14 | | 1367 | 21.35 | 46.21 | 46.54 |
| | | ViT-bigG-14 | | 2540 | 21.04 | 44.49 | 43.69 |
| | | xlm-roberta-base-ViT-B-32 | 5B | 366 | 16.79 | 37.49 | 40.91 |
| | | xlm-roberta-large-ViT-H-14 | | 1193 | 22.82 | 47.35 | 47.51 |
| | DataComp | small: ViT-B-32 | 13M | 151 | 12.06 | 22.90 | 21.19 |
| | | medium: ViT-B-32 | 128M | 151 | 16.95 | 28.28 | 33.01 |
| | | large: ViT-B-16 | 1B | 150 | 16.71 | 36.43 | 35.86 |
| | | xlarge: ViT-L-14 | 13B | 428 | 21.84 | 44.01 | 45.72 |
| SigLIP [85] | WebLI (English portion) | ViT-B | 13B | 172 | 24.29 | 48.57 | 49.06 |
| | | ViT-L | | 430 | 31.21 | 54.93 | 54.44 |
| | | ViT-SOViT | | 800 | **42.14** | **62.67** | **63.90** |

# C    Skill Analysis

We now provide the skill definitions and report model performance by each skill tag.

**Skill definitions and examples.** Table 8 provides definitions to the skills in NaturalBench.

**Skill analysis.** Table 9 reports **Q-Acc** performance (awarding one point if the model answers both images correctly for each question) on **Object** and **Attribute** tags. Table 10 reports **Q-Acc** performance on **Relation** and **Reasoning** tags.

**Additional examples.** We provide additional tagging examples in Figure 8. We will release these tags for more fine-grained analysis, such as evaluating models on combinations of skills.

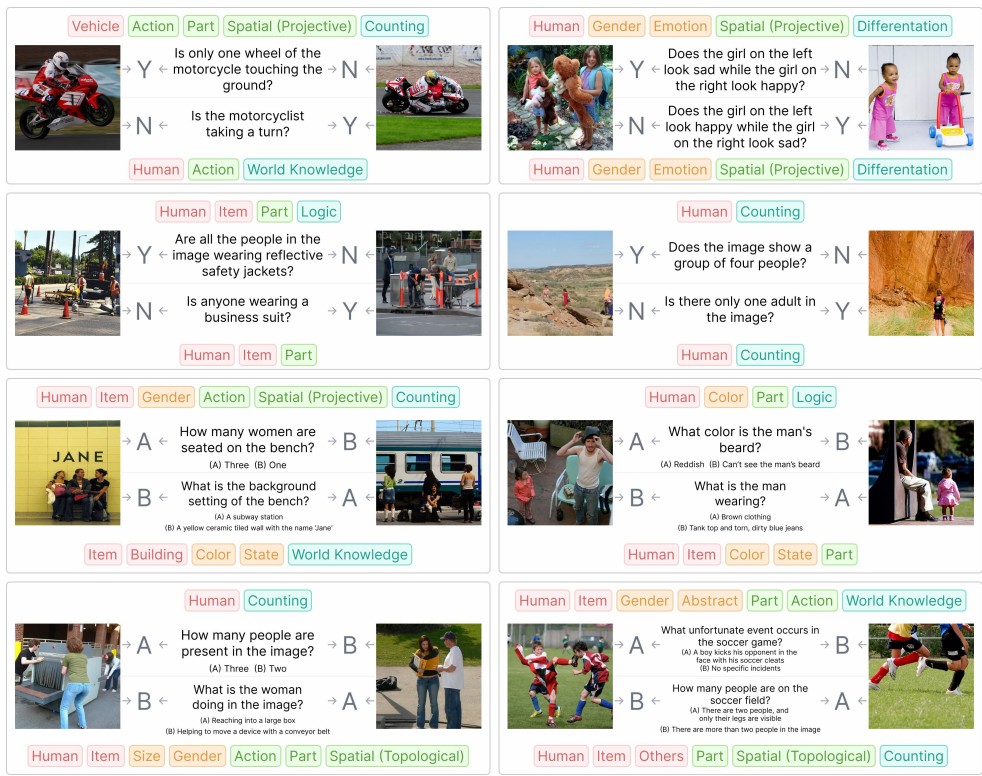

Figure 8: **More NaturalBench examples with skill tags.**

Table 8: **Skill definitions.**

| Skill Type | Definition | Examples |
|---|---|---|
| Object | Basic entities within an image, including animals, humans, food, buildings, natural elements (nature), vehicles, common items, and others. | *Is there a **car** parked near the path? Is there a **person** in this image? Is there a **referee** behind the **table**? Is the **dog** fully submerged in the **water** except for its **head**? Is the **water body** filled with visible **rocks** and emanating **ripples**?* |
| Attribute | Visual properties of entities, including emotion, shape, size, color, state, activity, gender, and abstract attributes (e.g., helpful, lucky). | *Is anyone in the picture **sad** or **scared**? Is the woman extremely **surprised**? Is the woman alone behind a **glass** partition? Is the man wearing **brown**? Is the man wearing a **red and white striped** apron? Is the **old** man in the image wearing **reflective** safety jackets?* |
| Spatial Relation | Physical arrangements of multiple entities relative to each other [46], including proximity (e.g., near, far), topological (e.g., at, on, in, with, surround, between, inside, outside), projective (e.g., left of, right of, under, in front of, below), orientation and direction (e.g., facing, towards, across, away from). | *Is there a referee **behind** the table? Is the dog looking **up** at the sky? Is there only one person **in** the canoe? Is there a group of people standing **outside** the gates? Is the man in the image looking **at** the object to his **left**? Is the smiling woman standing **next to** the bus?* |
| Action Relation | Action interactions between entities, e.g., pushing, kissing, hugging, hitting, helping, and so on. | *Is there a person **holding** a water bottle? Is the black dog **biting** a stick? Is anyone **using** an umbrella? Is the man **holding** a red pen? Is the dog **chasing after** a toy outdoors? Is the person **jumping directly off** a building without any equipment?* |
| Part Relation | Part-whole relationships between entities – one entity is a component of another, such as body part, clothing, and accessories. | *Is there a person wearing orange and yellow **shirt** and **jacket**? Is anyone wearing yellow and orange **safety vests**? Is the woman in the black **dress** wearing **gloves**? Is a player using **his back** to play the ball? Is the **boy's tongue** sticking out?* |
| Counting | Determining the quantity, size, or volume of entities, e.g., objects, attribute-object pairs, and object-relation-object triplets. | *Are there **four** people in the image? Does the dog have **two** visible colors? Are there **more than four** performers in the image?* |
| Differentiation | Differentiating objects within a category by their attributes or relations, such as distinguishing between "old" and "young" people by age, or "the cat on top of the table" versus "the cat under the table" by their spatial relations. | *Does the girl on the left look sad while the girl on the right look happy? Is there a cat sitting on a grey cabinet in front of another cat sitting on the stairs? Is one dog biting the ear of the other dog? Is a man standing behind another man sitting at a desk?* |
| Comparison | Comparing characteristics like number, attributes, area, or volume between entities. | *Does the scene involve players from three **different** team colors? Does the **tallest** building feature glass windows and side slopes? Is the **older** person following the **younger** one? Are there two dogs that are significantly **different** in size? Is the man wearing the **same** color as the woman in the image?* |
| Logic | Understanding logical operators. We only consider negation (as indicated by "no", "not", or "without") and universality (as indicated by "every", "all", "each", "both"). Other logical relations such as conjunction (as indicated by "and", "or") are omitted. | *Does the image show **all** men performing the same action? Are **both** people looking in the same direction? Is the bicycle rider performing a trick **without** any audience? Is the main subject **not** wearing shirt and lying down? Is the main activity potentially related to craft **or** construction?* |
| World Knowledge | Answering based on external commonsense knowledge, including social, symbolic, functional, physical, natural knowledge and so on. | *Is the event related to the Olympics? Is there a vertical depiction of Ramses III in the image? Does the image suggest a relatively informal social gathering? Is a single individual attempting to score regardless of multiple defenders?* |

Table 9: **Model performance on Object and Attribute.** We report **Q-Acc** on each tag.

| Model | Object | | | | | | | | Attribute | | | | | | | |
|---|---|---|---|---|---|---|---|---|---|---|---|---|---|---|---|---|
| | Animal | Human | Food | Building | Nature | Vehicle | Items | Others | Emotion | Shape | Size | Color | State | Abstract | Activity | Gender |
| BLIP-3(XGen-MM) | 18.6 | 16.2 | 15.4 | 20.8 | 21.7 | 22.2 | 21.2 | 17.6 | 9.1 | 19.3 | 24.1 | 21.8 | 20.2 | 20.4 | 16.5 | 14.0 |
| Phi-3-Vision | 15.6 | 17.1 | 15.4 | 17.7 | 15.6 | 19.0 | 18.5 | 16.7 | 18.2 | 17.5 | 19.0 | 18.9 | 16.8 | 15.6 | 15.2 | 15.8 |
| DeepSeek-VL-7B-Chat | 20.9 | 16.9 | 15.4 | 21.9 | 22.1 | 16.7 | 19.3 | 19.0 | 12.1 | 24.6 | 21.4 | 20.8 | 19.5 | 16.7 | 20.1 | 14.6 |
| LLaVA-NeXT(Mistral-7B) | 14.2 | 16.1 | 17.3 | 14.0 | 13.4 | 18.1 | 16.7 | 15.2 | 15.2 | 19.3 | 14.6 | 16.3 | 15.7 | 14.1 | 14.4 | 17.9 |
| InternLM-XC-V2-7B | 23.3 | 28.6 | 19.2 | 30.8 | 23.6 | 30.6 | 27.8 | 29.0 | 33.3 | 31.6 | 30.2 | 27.8 | 25.8 | 23.3 | 27.0 | 30.1 |
| InternVL-Chat-V1.2-Plus | 23.9 | 28.0 | 23.1 | 20.3 | 18.5 | 22.7 | 25.4 | 19.7 | 21.2 | 17.0 | 20.0 | 24.8 | 22.8 | 19.3 | 26.2 | 30.4 |
| GPT-4o | **35.4** | **39.7** | **44.2** | **40.1** | **41.3** | **38.4** | **42.8** | **38.3** | **39.4** | **42.1** | **40.7** | **39.0** | **41.1** | **38.9** | **35.5** | **43.2** |

Table 10: **Model performance on Relation and Reasoning.** We report **Q-Acc** on each tag.

| Model | Relation | | | | | | Reasoning | | | | |
|---|---|---|---|---|---|---|---|---|---|---|---|
| | Action | Part | Proximity | Topological | Projective | Orientation | Count | Logic | Differ | Compar | World |
| BLIP-3(XGen-MM) | 18.3 | 17.4 | 27.5 | 22.8 | 19.6 | 15.5 | 20.6 | 15.9 | 13.0 | 20.9 | 5.3 |
| Phi-3-Vision | 16.0 | 19.5 | 19.6 | 17.9 | 13.9 | 9.5 | 16.1 | 18.5 | 17.6 | 13.0 | 8.5 |
| DeepSeek-VL-7B-Chat | 17.5 | 16.2 | 29.4 | 21.4 | 17.9 | 14.7 | 19.6 | 16.4 | 11.1 | 11.3 | 10.6 |
| LLaVA-NeXT(Mistral-7B) | 15.9 | 18.6 | 18.6 | 17.0 | 16.1 | 13.8 | 17.1 | 21.2 | 17.6 | 12.2 | 9.6 |
| InternLM-XC-V2-7B | 27.3 | 29.3 | 29.4 | 27.9 | 24.4 | 24.1 | 30.7 | 25.9 | 27.8 | 27.8 | 17.0 |
| InternVL-Chat-V1.2-Plus | 23.6 | 28.1 | 31.4 | 24.4 | 19.3 | 18.1 | 23.9 | 26.9 | 25.0 | 15.7 | 12.8 |
| GPT-4o | **39.4** | **43.1** | **40.2** | **41.7** | **38.7** | **35.3** | **39.2** | **42.9** | **38.9** | **37.4** | **35.1** |

# D  Debiasing Analysis

In the main paper, we show that debiasing within the image-text pairings significantly improves model performance. Here, we explore debiasing techniques that don't rely on knowing the image-question pairings.

**Deterministic evaluation using answer likelihood [44].** Recall that we can perform a scoring-based evaluation strategy using the generative likelihood of each candidate answer (VQAScore [44]) to determine the model's predicted answer. Specifically, given a question $q$, an image $i$, and two candidate answers $a_0$ and $a_1$, we evaluate:

$$P(a_0|q,i) - P(a_1|q,i) > \tau \tag{4}$$

where $\tau$ is a threshold (default is 0). If this condition (Eq. 4) is met, the model predicts $a_0$; otherwise, it predicts $a_1$. Crucially, this formulation has two benefits: (1) it produces deterministic results that are almost consistent with stochastic decoding (see Table 11), and (2) it allows us to adjust $\tau \in [-1, 1]$ for debiasing. Recall that our main paper performs **sample**-level debiasing by optimizing $\tau$ within each of the four image-question pairs. Alternatively, we can perform **global**-level debiasing by searching for a single $\tau$ that maximizes **G-Acc** across all samples. We also implement the **post-hoc** debiasing technique proposed in [88], which is equivalent to:

$$\frac{P(a_0|q,i)}{P(a_0|q)} - \frac{P(a_1|q,i)}{P(a_1|q)} > 0 \tag{5}$$

where $P(a|q)$ is estimated by sending no image tokens but just the question tokens to the VLM. Table 11 shows that these alternate techniques still lag behind the performance of sample-level debiasing. We hope NaturalBench can be a useful testbed for bias mitigation techniques for VLMs.

Table 11: **Evaluating debiasing techniques on NaturalBench.** We evaluate debiasing techniques (as detailed in Section 5) that do not require prior knowledge of image-question pairings (unlike sample optimal $\tau$). For comprehensiveness, we report both stochastic decoding and deterministic evaluation using VQAScore, finding consistent results. We observe that the two post-hoc methods – global-optimal $\tau$ and Post-Hoc debiasing – perform significantly worse than the (oracle) sample-optimal $\tau$. Global optimal $\tau$ shows only slight improvements, while Post-Hoc debiasing even reduces performance in models like Bunny, InterVL-Chat-V1.2, and GPT-4o. This suggests NaturalBench can be a valuable benchmark for testing future debiasing methods.

| Model | Stochastic Decoding | | | Deterministic VQAScore | | | Post-hoc Debiasing [88] | | | Global Optimal $\tau$ | | | Sample Optimal $\tau$ | | |
|---|---|---|---|---|---|---|---|---|---|---|---|---|---|---|---|
| | Q-Acc | I-Acc | G-Acc | Q-Acc | I-Acc | G-Acc | Q-Acc | I-Acc | G-Acc | Q-Acc | I-Acc | G-Acc | Q-Acc | I-Acc | G-Acc |
| LLaVA-1.5 (Vicuna-7B) | 37.7 | 43.8 | 12.7 | 36.7 | 42.7 | 12.2 | 38.2 | 44.5 | 13.9 | 39.9 | 45.8 | 14.0 | 83.4 | 76.3 | 44.3 |
| LLaVA-1.5 (Vicuna-13B) | 39.6 | 44.6 | 14.8 | 38.6 | 43.5 | 14.4 | 38.5 | 42.8 | 14.5 | 42.8 | 47.8 | 16.5 | 86.2 | 78.6 | 49.7 |
| Phi3-Vision | 43.4 | 48.7 | 17.2 | 43.6 | 48.9 | 17.7 | 45.1 | 48.6 | 19.3 | 44.7 | 49.3 | 18.4 | 85.7 | 78.5 | 50.0 |
| Bunny | 42.3 | 48.4 | 17.4 | 42.5 | 48.5 | 17.5 | 38.7 | 44.9 | 15.7 | 43.6 | 49.5 | 18.7 | 85.8 | 78.6 | 50.5 |
| LLaVA-NeXT (Vicuna-7B) | 42.5 | 47.6 | 15.0 | 42.0 | 47.1 | 15.0 | 44.2 | 48.9 | 18.0 | 43.4 | 48.5 | 16.5 | 86.7 | 79.6 | 50.3 |
| LLaVA-NeXT (Mistral-7B) | 44.6 | 49.1 | 16.3 | 45.0 | 49.4 | 17.0 | 46.8 | 51.1 | 19.6 | 45.3 | 49.7 | 17.4 | 88.3 | 81.6 | 56.0 |
| LLaVA-NeXT (Vicuna-13B) | 45.9 | 49.9 | 19.2 | 44.6 | 48.5 | 18.2 | 48.7 | 52.5 | 21.5 | 47.8 | 52.1 | 20.4 | 89.1 | 82.3 | 57.2 |
| DeepSeek-VL-7B-Chat | 46.0 | 50.1 | 19.3 | 45.8 | 49.9 | 19.4 | - | - | - | 46.4 | 50.4 | 19.7 | 86.6 | 81.8 | 54.8 |
| BLIP-3(XGen-MM) | 47.0 | 51.2 | 19.5 | 46.8 | 51.1 | 19.5 | 47.8 | 52.0 | 22.4 | 48.7 | 53.2 | 21.4 | 88.6 | 81.9 | 55.3 |
| InternVL-Chat-V1.5 | 52.3 | 55.9 | 23.1 | 52.6 | 56.0 | 24.3 | 55.2 | 58.4 | 28.6 | 52.3 | 55.6 | 25.0 | 92.3 | 86.1 | 66.0 |
| InterVL-Chat-V1.2-Plus | 52.7 | 56.2 | 23.4 | 52.6 | 56.3 | 23.5 | 55.9 | 58.6 | 28.3 | 53.0 | 56.1 | 24.6 | 92.4 | 85.5 | 65.3 |
| InternVL2-8B | 50.5 | 54.5 | 23.6 | 50.4 | 54.3 | 23.7 | 52.2 | 55.9 | 25.5 | 50.4 | 54.3 | 23.7 | 88.7 | 83.2 | 58.6 |
| InternVL-Chat-V1.2 | 52.9 | 56.4 | 26.6 | 52.6 | 56.0 | 26.2 | 52.3 | 54.3 | 25.8 | 53.6 | 56.8 | 27.2 | 91.6 | 86.0 | 65.8 |
| InternVL2-26B | 55.9 | 58.8 | 28.1 | 55.7 | 58.5 | 28.2 | 58.8 | 61.1 | 32.0 | 55.7 | 58.3 | 28.5 | 92.2 | 87.2 | 67.7 |
| LLaVA-OneVision (Qwen2-0.5B) | 39.8 | 46.3 | 15.7 | 39.1 | 44.6 | 14.5 | 39.1 | 44.5 | 15.8 | 39.2 | 46.3 | 16.2 | 84.6 | 77.2 | 47.5 |
| LLaVA-OneVision (Qwen2-7B) | 56.2 | 58.8 | 28.9 | 55.4 | 58.2 | 28.6 | 59.1 | 61.2 | 33.2 | 56.1 | 59.0 | 28.7 | 92.1 | 87.2 | 67.8 |
| GPT-4o | **64.4** | **66.4** | **39.6** | **65.0** | **67.0** | **40.5** | **61.6** | **63.2** | **37.6** | **64.9** | **67.1** | **40.7** | **94.0** | **90.5** | **75.6** |

