# OpenReview forum: "NaturalBench: Evaluating Vision-Language Models on Natural Adversarial Samples"
_NeurIPS.cc/2024/Datasets_and_Benchmarks_Track — NeurIPS 2024 Track Datasets and Benchmarks Poster_

### Official Review · Reviewer_E37r · 2024-07-24

**Rating:** 7
**Confidence:** 4
**Clarity:** The paper is well written and easy to…

**Review:**

The paper is well-written and very relevant to recent developments in VLMs. It addresses major pain points of current vision-language benchmarks and is a valuable contribution

Strengths
- The benchmark is well motivated
- Uses real images, instead of synthetically generated ones like prior works
- Use a thorough annotation protocol, requiring high levels of inter-annotator agreement
- Show that the dataset is robust to blind solutions
- Show that there is a big gap between frontier models and human performance
- Presents a thorough analysis of model performance, including many models and data breakdowns

Weaknesses:
- It seems like the performance when using deterministic vs stochastic is not fully consistent. What is the reason for that? Is it e.g. worse models not following instructions, and so the answers they return can not be properly parsed?
- The paper says thes code and data is on a website, but the url to the website is not given (I could also not find it online)
- The comparison to human performance might not be entirely fair. When evaluating human performance, it could be that humans have already seen an image or question at a previous time (because each image and each question would appear twice), which can bias them to select a different answer this time round as they know answers would not repeat. First, was any care taken to prevent this? Second, would a VLM that can process interleaved images and text show better performance if it has previous answers in its context? (The latter is partially answered in Section 5, albeit in an easier scenario, where the model is forced to select different answers – this significantly increases performance and approaches that of humans)
- Dynamic evaluation: Section 6 briefly describes using additional data sources. Were humans used again to filter the questions? What is the quality of this automatically generated dataset? This is important in orderto claim this is a dynamic benchmark

**Strengths:**

See review

**Additional Feedback:**

-

**Correctness:**

The dataset collection and evaluation look good. I have a slight concern about fairness when comparing to humans and automatic labelling as described in Section 6 (see Review)

**Documentation:**

There is no URL for accessing the dataset as far as I can see

**Ethics:**

No concerns

**Limitations:**

The limitation section mentions that human verification is crucial for detecting potential biases in the data, but the human verification done in this work does little to tackle bias.

**Opportunities For Improvement:**

See review

**Relation To Prior Work:**

Yes

**Summary And Contributions:**

This paper proposes NaturalBench, consisting of images and captions that form “natural adversarial examples” – visual questions that human easily answer, but VLMs struggle with. The paper uses off-the-shelf models to automatically mine text-image pairs that are hard for VLMs, where each evaluation sample consists of two images and two questions, where each question has a different answer for each image. This framework of balanced answers and automatic mining solves several problems in current benchmarks, like i) blind models performing better than average, iii) data leakage into the training set of VLMs and iii) model performance saturation

---

> ### Author Rebuttal · Authors · 2024-08-16
>
> We sincerely appreciate your positive and constructive feedback. We address your concerns below:
>
> > **It seems like the performance when using deterministic vs stochastic is not fully consistent. What is the reason for that? Is it e.g. worse models not following instructions, and so the answers they return can not be properly parsed?**
>
> Yes, your intuition is correct. Many models do not consistently follow instructions, leading to slightly lower stochastic performance (0.2 to 2.0 points) compared to deterministic performance. We will clarify this in the revised manuscript.
>
> > **The paper says thes code and data is on a website, but the url to the website is not given (I could also not find it online)**
>
> We apologize for the omission of the URLs. The NaturalBench dataset is now hosted on Huggingface:
>
> https://huggingface.co/datasets/BaiqiL/NaturalBench
>
> We will release the code for NaturalBench evaluation (and other benchmark evaluations) at:
>
> https://github.com/Nyandwi/naturalbench
>
> > **The comparison to human performance might not be entirely fair. When evaluating human performance, it could be that humans have already seen an image or question at a previous time (because each image and each question would appear twice), which can bias them to select a different answer this time round as they know answers would not repeat. First, was any care taken to prevent this?**
>
> Thank you for raising this important question. We prevent potential bias by ensuring each annotator sees only one image-question pair from each sample (which consists of four pairs). This is implemented during both the (1) human verification and (2) testing phases:
>
> - **Human verification**: During collection, each annotator sees only one image-question pair per sample. Additionally, each pair must be answered correctly by at least two annotators, or it is discarded. We employ a pool of 8 annotators for this process.
> - **Human testing**: For testing, a separate group of 8 annotators is used, with each assigned only one image-question pair per sample to prevent cheating. An image-question pair is considered correct only if both annotators answer it correctly.
>
> We will add this clarification to the revised manuscript. Thank you for pointing this out!
>
> > **Second, would a VLM that can process interleaved images and text show better performance if it has previous answers in its context? (The latter is partially answered in Section 5, albeit in an easier scenario, where the model is forced to select different answers – this significantly increases performance and approaches that of humans)**
>
> This is an excellent point. As you suggested, requiring models to select a different answer for each image (or question) can significantly boost performance, bringing leading models like GPT-4o close to human-level accuracy. Using interleaved in-context image-text samples could be another promising approach for debiasing VLMs and merits further research.
>
> > **Dynamic evaluation: Section 6 briefly describes using additional data sources. Were humans used again to filter the questions? What is the quality of this automatically generated dataset? This is important in orderto claim this is a dynamic benchmark**
>
> Yes, we used the same human verification procedure to filter the generated questions from the DOCCI and XM3600 datasets. In addition, for the multi-lingual VQA benchmarks from XM3600, we focused on Chinese and Hindi, as the authors are native speakers of these languages. We will clarify this in the revised manuscript.

---

> > ### Comment · Reviewer_E37r · 2024-08-27
> > **Thanks for the response**
> >
> > Thank you for the response; I will keep my original rating (7 - accept)

---

### Official Review · Reviewer_4Nfw · 2024-07-24
**A well-designed, impactful, and practical vision-language evaluation benchmark**

**Rating:** 9
**Confidence:** 4
**Correctness:** Yes
**Clarity:** Yes

**Review:**

N/A

**Strengths:**

- Address issues like knowledge prior in the vision-language model evaluation, and propose a fair, efficient, dynamic benchmark to avoid these issues.
- Analyze why the benchmark is challenging from compositionality and biases. The authors also tag data samples with multiple associated skills to show the compositional skills.
- The authors widely evaluate state-of-the-art open-source and closed-source vision-language models on the proposed benchmark.

**Additional Feedback:**

N/A

**Documentation:**

Yes

**Opportunities For Improvement:**

It’s a very good work for me. It is better to demonstrate the effectiveness of selecting mismatched image caption pairs.

**Relation To Prior Work:**

Yes

**Summary And Contributions:**

This paper proposes a semi-automatic framework to collect natural adversarial samples for VLM evaluation, as well as a benchmark that evaluates image-question pairs to avoid blind solutions, and points out why this benchmark is hard for most vision-language models. To construct the dataset, the authors first filter a set of image-text pairs that mismatched by models like CLIP, and then use ChatGPT to generate questions for the image pairs.

---

> ### Author Rebuttal · Authors · 2024-08-16
>
> We are glad that you like our work! We address your concern below:
>
> > **It is better to demonstrate the effectiveness of selecting mismatched image caption pairs.**
>
> This is a great suggestion. Our Appendix Table 7 shows that using mismatched image-caption pairs creates a significantly more challenging VQA benchmark, with most models, including GPT-4o, experiencing a performance drop of more than 50% in G-Acc. We will consider moving this table to the main paper in the revised manuscript.

---

> > ### Comment · Reviewer_4Nfw · 2024-08-30
> > **Thanks for the reply**
> >
> > Thanks for updating the paper!

---

### Official Review · Reviewer_VBjr · 2024-07-25
**NaturalBench: Evaluating Vision-Language Models on Natural Adversarial Samples**

**Rating:** 6
**Confidence:** 4
**Correctness:** Yes
**Clarity:** Yes

**Review:**

NaturalBench is a good contribution to the evaluation of VLMs, providing a robust dataset that challenges models with natural adversarial examples rather than synthetic or artificially constructed ones.
The semi-automated method for generating the benchmark using off-the-shelf models like CLIP and ChatGPT reduces the manual effort required for dataset creation and ensures scalability.

**Strengths:**

The benchmark includes a diverse range of VQA samples, each tagged with multiple visio-linguistic skills, allowing for a detailed analysis of model capabilities across various dimensions such as attribute bindings, object relationships, and advanced reasoning.
The evaluation protocol prevents "blind" solutions by pairing each question with two images that require different answers, ensuring models must rely on visual inputs rather than language priors.

**Additional Feedback:**

N/A

**Documentation:**

Yes

**Limitations:**

The introduction of new evaluation metrics like question accuracy (Q-Acc), image accuracy (I-Acc), and group accuracy (G-Acc) provides a nuanced view of model performance but might complicate the comparison with existing benchmarks that use more straightforward metrics.

**Opportunities For Improvement:**

While NaturalBench aims to provide a balanced evaluation, the reliance on natural image-text corpora might introduce inherent biases from these datasets. The paper acknowledges this but does not extensively discuss strategies to mitigate such biases.

**Relation To Prior Work:**

Yes

**Summary And Contributions:**

This paper introduces NaturalBench, a benchmark designed to evaluate the robustness of VLMs against natural adversarial samples. These samples are naturally occurring image-text pairs that challenge the model's visual reasoning capabilities, unlike artificially crafted adversarial examples. The benchmark comprises over 10,000 human-verified VQA samples, making it one of the most comprehensive benchmarks in the field. The authors demonstrate that existing VLMs, including state-of-the-art models like GPT-4, struggle significantly with these adversarial samples, performing only marginally better than random guessing.

---

> ### Author Rebuttal · Authors · 2024-08-16
>
> We sincerely appreciate your positive and constructive feedback. We address your concerns below:
>
> > **While NaturalBench aims to provide a balanced evaluation, the reliance on natural image-text corpora might introduce inherent biases from these datasets. The paper acknowledges this but does not extensively discuss strategies to mitigate such biases.**
>
> This is a great question. We believe that *dynamic evaluation* using new image-text corpora, as discussed in Section 6, can help mitigate biases from relying on a single dataset like Flickr. While Flickr focuses mostly on human activities, the recently released DOCCI [1] dataset emphasizes skills that Flickr lacks, such as *Natural Elements* (for scene understanding) and *Symbols* (for optical character recognition), which are five times more prevalent in DOCCI. We will provide a detailed skill distribution for each dataset in the revised manuscript.
>
> With our semi-automated data curation protocol, researchers can easily repurpose new image-text corpora with varying skill distributions into robust VQA benchmarks.
>
> > **The introduction of new evaluation metrics like question accuracy (Q-Acc), image accuracy (I-Acc), and group accuracy (G-Acc) provides a nuanced view of model performance but might complicate the comparison with existing benchmarks that use more straightforward metrics.**
>
> We agree that for clarity, we should report the standard binary VQA accuracy, and we do so in Table 1.
>
> As you suggested, although Q-Acc, I-Acc, and G-Acc are somewhat more complex, they offer deeper insights into model biases. For example, Table 2 shows that enforcing the model to select different answers for two images of the same question (i.e., debiased Q-Acc) can significantly raise the Q-Acc of the state-of-the-art GPT-4o to human levels. This suggests that mitigating answer biases in existing models could be a promising avenue for future improvement.
>
> **References**:
>
> [1] Onoe et al. DOCCI: Descriptions of Connected and Contrasting Images. ECCV 2024.

---

### Comment · Area_Chair_Cr1B · 2024-08-27
**Request for Update**

Dear Reviewers,

As we approach the end of the discussion phase, I would like to kindly remind you to review the author’s rebuttal to see if it addresses the concerns you’ve raised. Your input is crucial in ensuring a thorough and fair evaluation, so if the rebuttal clarifies or resolves any of your concerns, please consider updating your comments accordingly. If your concerns persist, a follow-up comment would also be greatly appreciated.

Thank you for your continued dedication and contributions to the review process.

Best,

AC

---

### Decision · Program_Chairs · 2024-09-26

**Decision:**

Accept (Poster)

**Comment:**

All reviewers have provided positive scores for this submission, highlighting its strengths in novelty and contributions to the evaluation of VLMs. Given the unanimous positive feedback and the recognition of its contribution to the area, the AC carefully reviewed the paper and concurred with the reviewers' assessments, therefore supporting the decision to accept this submission.